# The Equine Dental Pulp: Analysis of the Stratigraphic Arrangement of the Equine Dental Pulp in Incisors and Cheek Teeth

**DOI:** 10.3390/vetsci9110602

**Published:** 2022-10-30

**Authors:** Jessica Roßgardt, Laura Beate Heilen, Kathrin Büttner, Jutta Dern-Wieloch, Jörg Vogelsberg, Carsten Staszyk

**Affiliations:** 1Institute of Veterinary-Anatomy, -Histology and -Embryology, Faculty of Veterinary Medicine, Justus-Liebig-University Giessen, Frankfurter Str. 98, 35390 Giessen, Germany; 2Unit for Biomathematics and Data Processing, Faculty of Veterinary Medicine, Justus-Liebig-University Giessen, Frankfurter Str. 95, 35390 Giessen, Germany

**Keywords:** equine dental pulp, subodontoblastic zone, capillary network, nerve fiber plexus, fibroblastic cells

## Abstract

**Simple Summary:**

The crown pulp of permanent brachydont (short-crowned) teeth features a stratigraphic arrangement composed of a cell-free and a cell-rich zone. This stratigraphy is assumed to reflect a mature, less active status with only minor production of secondary dentin. In contrast, the hypsodont (high-crowned) equine tooth is required to sustain a production of large amounts of subocclusal dentin to compensate for lifelong occlusal wear of several mm per year. It is hypothesized that the remarkable productivity of the equine pulp is reflected by distinct histomorphological features of the pulpal tissue. Therefore, teeth of horses of different ages were investigated, paying special attention to identify the cellular arrangement at the pulp–dentin interface. In general, the layer of odontoblasts was followed by a dense arrangement of fibroblastic cells in a network of small-sized blood vessels as well as a high concentration of single nerve fibers, suggesting a subodontoblastic supportive zone. The formation of distinct zones as described in the brachydont crown pulp is absent. The observed histomorphological characteristics of the equine pulp are regarded as representing a highly productive status, which presumably accounts for the reception of mechanical stimuli; a high capacity for cellular regeneration; and finally, the required subocclusal dentin production.

**Abstract:**

In the crown pulp of brachydont teeth, a cell-free and a cell-rich zone are established beneath the odontoblastic layer, indicating a mature status. For the equine dental pulp, there are no descriptions which allow for a comparative analysis with regard to functional requirements in terms of lifelong secondary dentin production to compensate for occlusal wear. For histomorphological and immunohistological investigations, ten incisors and ten check teeth were used from seven adult horses and five foals. In the periphery of the equine dental pulp, a constant predentin and odontoblastic cell layer was present, followed by densely packed fibroblastic cells, capillary networks, and a high concentration of nerve fibers, suggesting a subodontoblastic supportive zone. Whilst the size of the equine dental pulp decreased with age, the numbers of blood vessels, nerve fibers, and fibroblastic cells increased with age. Histological analysis of the equine dental pulp did not show a cell-free and cell-rich zone as described in the brachydont crown pulp. The equine dental pulp remained in a juvenile status even in aged horses, with morphological features indicating a high capacity for dentine production.

## 1. Introduction

Equine hypsodont teeth, incisors, and cheek teeth feature a complex pulp cavity system underneath their occlusal surfaces [1,2]. To seal the occlusal part of the dental pulp and to prevent pulpal exposure due to massive dental wear, a compensating production of a special type of dentin is necessary, the so-called irregular secondary dentin [3,4]. Its lifelong production is maintained by the functional cells of the pulpal tissue, the odontoblasts [5,6]. Although brachydont teeth of dogs and humans also feature a continued production of dentin, the extent of subocclusal dentin production in the horse exceeds that of brachydont teeth by far [7]. Therefore, the question arises whether these functional differences are also reflected by morphological differences. 

Within the coronal pulp of mature human brachydont teeth, the palisade-pattern-like odontoblastic layer is located in the most peripheral part of the dental pulp, adherent to the predentin layer [8,9,10]. In the central direction, the odontoblastic layer is lined by a cell-free zone (also referred to as Zone of Weil) [10,11,12], in which individual, non-myelinated nerve fibers as well as cellular processes of bipolar fibroblasts are present [13]. The cell-free zone is followed by a nucleus-rich zone, termed cell-rich zone, housing densely packed fibroblasts responsible for the production of a supporting collagen fiber network [14]; undifferentiated mesenchymal cells; and cells of the immune system, including lymphocytes, granulocytes, and macrophages [15,16,17]. Furthermore, a subodontoblastic capillary plexus as well as a high concentration of nerve fibers have been recorded [10,18]. The subodontoblastic nerve fibers form a plexus, referred to as Raschkow plexus [19,20]. Isolated capillaries as well as nerve fibers from the Raschkow plexus branch out towards the odontoblastic layer and form plexus-like arrangements between odontoblasts, potentially reaching out into the dentinal tubules [21,22,23]. Although data from investigations in brachydont teeth of non-human species are rarely available, it is assumed that such a stratigraphic microscopic morphology is generally established in the crown pulp of brachydont teeth [8,11]. The described stratigraphic organization of the brachydont crown pulp is absent in the brachydont root pulp [9,20] as well as in immature brachydont teeth, suggesting that the observed stratigraphic organization indicates a mature and less active status of the pulp [12,14]. 

Hitherto, no detailed microscopical descriptions of the pulp of equine teeth are available, which would allow for a comparative analysis of the pulpal morphology with regard to the special cellular requirements of the equine hypsodont tooth. Numerous literature exists regarding histological characteristics in brachydont teeth and even in the hypselodont and enigmatic narwhal tooth a plexus-like arrangement of nerve fibers is observed [24]. Therefore, the aim of this study was to elucidate the cellular, neuronal, and vascular details of the equine dental pulp, focusing on the presence of stratigraphic arrangements. These data might be utilized for the development of equine specific dental and endodontic therapies.

## 2. Materials and Methods

### 2.1. Sampling

Maxillary cheek teeth (n = 10, Triadan 08) and mandibular incisors (n = 10, Triadan 01) [25,26] were examined from five foals and seven adult horses, which had been euthanized for other reasons than dental diseases (see Table 1). Before further processing, the horses were assigned into three age groups (AG):

AG 1: 0 to 210 days;

AG 2: 5 to 14 years;

AG 3: 19 to 24 years.

### 2.2. Gross Anatomical Examination and Sectioning

All samples were treated according to Roßgardt et al. [27]. In brief, incisor and cheek teeth rows were dissected by a band saw (K440 H, Kolbe Foodtec, Elchingen, Germany), and all samples were stored in formalin (10%, pH 7, 4 °C). All teeth were inspected macroscopically with regard to clinical health. Teeth displaying any signs of dental or periodontal diseases (e.g., tooth fractures, signs of pulpitis, periodontal pockets) were excluded from this study. Three defined horizontal sections were obtained, utilizing a diamond-coated micro-band saw (MBS 240/E, Proxxon S.A., Wecker, Luxembourg):

Subocclusal (so);

Central (c);

Apical (a).

### 2.3. Staining and Immunohistochemistry

After decalcification in buffered EDTA solution (ethylenediaminetetraacetate, pH 8, 20 °C), the samples were flushed and stored overnight in PBS (phosphate-buffered saline, Carl Roth GmbH, Karlsruhe, Germany). The final samples were embedded in paraffin wax (automatic embedding system, EG1150h, Leica Biosystems, Nussloch GmbH, Nussloch, Germany), and 7 µm histological sections were prepared and stained with toluidine blue (for further details see Roßgardt et al. [25]). A subset of consecutive sections was used for immunohistochemical staining. Deparaffinization, using a descending alcohol series, was followed by enzymatic treatment with proteinase K and blocking of endogenous peroxidase. To prevent non-specific background staining, blocking buffer (buffer solution and bovine serum albumin, Carl Roth GmbH, Karlsruhe, Germany) was used. For the detection of blood vascular endothelial cells, primary antibodies against von-Willebrand factor VIII (vWF) (polyclonal rabbit, anti-human, A0082, dilution 1:300, Dako Denmarkt A/S, Glostrup, Denmark) were added. To detect neuronal structures, primary antibodies against neurofilament proteins (NF) (monoclonal mouse anti-human, M0726, dilution 1:50, Dako Denmark A/S, Glostrup, Denmark) were applied.

After storage overnight at room temperature, primary antibodies were washed off using PBS. Subsequently, peroxidase-conjugated secondary antibodies (peroxidase-conjugated AffiniPure goat anti rabbit/mouse IgG, Jackson ImmunoResearch Europe Ltd, Bar Harbor, ME, USA) were added, and immune reactions were visualized using AEC (amino ethyl carbazole, substrate kit SK4200, Vector Laboratories, Newark, CA, USA) as a chromogen. Serial sections of equine mandibular lymph nodes and equine parotid and sublingual glands were used as positive controls. Negative controls were generated by omitting the primary antibodies.

### 2.4. Histological Evaluations

Stained sections were investigated by light microscopy (Leica DM2500, Leica Microsystems GmbH, Wetzlar, Germany) applying an automated image building tool (10 or 20× magnification, Leica LAS XY Live Image Builder, Leica Microsystems GmbH, Wetzlar, Germany). A radial relative homogeneous distribution of tissue component and cell types within the pulp horn was verified by preliminary analysis of 50 pulp horns of different aged teeth on all horizontal sections (subocclusal, central, and apical). Thus, further histomorphometrical data were obtained from four zones of constant length (100 µm) and recalculated width depending on the maximum width of the pulp. These zones were set adjacent to the odontoblastic layer. Additionally, two zones, 1.a and 1.b of constant length (100 µm) and width (20 µm) were defined in the periphery of the pulp area to clarify whether there is a stratigraphic arrangement (see Figure 1).

Within the defined zones, the number of blood vessels and nerve fibers were determined manually, using the immunohistochemically stained sections for blood endothelial cells and neuronal fibers. Toluidine blue-stained sections were used for counting fibroblastic cell nuclei within the defined zones with an automated analyzing tool (Analysis, Measurements, LAS X, Leica Microsystems GmbH, Wetzlar, Germany). Furthermore, the diameter of the blood vessels was evaluated (<10 µm, 10–50 µm, >50 µm) by a lining tool (Quantify, Draw Line, Leica LAS X, Leica Microsystems GmbH, Wetzlar, Germany). The density of blood vessels, nerve fibers, and fibroblastic cell nuclei was defined as number per 2000 µm^2^.

### 2.5. Statistical Analysis

To illustrate the descriptive statistics, GraphPad Prism 6 (GraphPad Software, Inc., La Jolla, CA, USA) was used. The inferential statistics included in the charts were initially analyzed by SigmaStat 4.0 (Systat Softwares Inc., San José, CA, USA). All data sets were charted as described in Roßgardt et al. [27]. Methods of SigmaStat 4.0 were implemented to calculate the inferential statistics. Therefore, the evaluation of the density of blood vessels, nerve fibers, and fibroblastic cell nuclei was performed by a two-way analysis of variance (ANOVA) with repeated measurements. This repetition of measurements was reached by various section planes of one tooth. One factor was represented by the age group, AG 1–3; the second was represented by the subocclusal, central, or apical sectional planes; and the third was represented by the zones adjacent to the subodontoblastic layer (zone 1.a, zone 1.b, and zone 1–4). Initially, a two-way ANOVA was performed separately within each age group, concerning the two factors section plane and zone. Subsequently, another two-way ANOVA was performed within each zone for the two fixed factors age group and section plane. Providing that significances in the variables, density of blood vessels, nerve fibers, and fibroblastic cell nuclei could be detected, and further analyses were obtained by applying the post hoc test by Tukey. When normal distribution was lacking, a logarithmic transformation, an inverse transformation, or a Johnson transformation was applied. In three cases, the normal distribution was still missing, and thus, a one-way ANOVA was implemented. Therefore, the data were converted from indexed variables to unindexed ones to obtain the individual groups for each combination of the factors age group and section plane. If the values still were not normally distributed, which was the case once, a Friedman test was performed. Generally, the significance level was determined at α ≤ 0.05. In the following diagrams, *p*-values are indicated as *p* ≤ 0.05 (*), *p* ≤ 0.01 (**), and *p* ≤ 0.001 (***).

## 3. Results

### 3.1. Blood Vessels

In incisors as well as in cheek teeth, the density of blood vessels per 2000 µm^2^, measuring a diameter < 10 µm and a diameter between 10–50 µm, showed an increasing trend from AG 1 to AG 3. No statistical significance was obtained, except for blood vessels with a diameter < 10 µm in cheek teeth (within zone 2: AG 1 vs. 2: *p* = 0.02; AG 1 vs. 3: *p* = 0.002; see Figure 2 and Appendix A). Regarding the defined horizontal subocclusal (*so*), central (*c*), and apical (*a*) sections, no statistically significant differences were recorded, and thus, the different levels were combined in each age group.

In incisors, the highest density of blood vessels with a diameter < 10 µm was obtained in zone 1.a next to the odontoblastic layer, showing increasing absolute values from AG 1 (mean (m) = 0.83/2000 µm^2^) to AG 2 (m = 1.33/2000 µm^2^) to AG 3 (m = 1.88/2000 µm^2^). Regarding the different assessed zones, the density of blood vessels with a diameter < 10 µm in all age groups decreased significantly from zone 1.a to zone 4, i.e., from peripheral to central (see Figure 3). In contrast, blood vessels with a diameter measuring 10–50 µm showed a significant increase in the central direction (AG 1: zone 1.a and 1.b vs. zone 2: *p* = 0.008; AG 2: zone 1.a and 1.b vs. zone 2: *p* ≤ 0.001), with highest absolute values in zone 2 (AG 1: m = 0.2/2000 µm^2^; AG 2: m = 0.33/2000 µm^2^; AG 3: m = 0.41/2000 µm^2^). In zones 1.a and 1.b, no blood vessels with a diameter above 10 µm were detected, except for AG 3 (m = 0.22/2000 µm^2^).

In cheek teeth, the findings for blood vessels < 10 µm per 2000 µm^2^ were similar to those in incisors regarding the assessed zones. High values were detected in the peripheral zones 1.a and 1.b and in zone 1, showing a decrease from zone 1 to zone 4. In general, the amount of blood vessels with a diameter < 10 µm in cheek teeth was higher than that in incisors, with absolute highest values in zone 1.b for AG 2 (m = 2.36/2000 µm^2^) and AG 3 (m = 3.81/2000 µm^2^). Similar to the incisors, the number of blood vessels with a diameter measuring 10–50 µm significantly increased from zone 1.a to zone 4 (AG 1: zone 1.a and 1.b vs. zone 3: *p* = 0.021; AG 2: zone 1.a vs. zone 3 and 4: *p* ≤ 0.05). The density of blood vessels with a diameter of 10–50 µm obtained in zones 2, 3, and 4 remained nearly constant in each age group. If blood vessels with a diameter > 50 µm were detected, they were found in zones 3 and 4, i.e., in the center of the pulp, showing no statistical significances.

### 3.2. Nerve Fibers

The density of nerve fibers in incisors as well as in cheek teeth showed a significant increase from AG 1 to AG 3 (incisors: within zone 1.a: AG 1 vs. AG 3: *p* = 0.003; cheek teeth: within zone 3: AG 1 vs. AG 3: *p* < 0.001; AG 2 vs. AG 3: *p* < 0.001; see Figure 3, Figure 4 and Figure 5 and Appendix A). In deciduous teeth of foals (AG 1), the density of nerve fibers remained constantly low (i.e., < 3 nerve fibers/2000 µm^2^) in incisors as well as in cheek teeth, with highest values in the subocclusal level in the peripheral zone 1.a (incisors: m = 0.67/2000 µm^2^; cheek teeth: m = 2.67/2000 µm^2^). The density of nerve fibers in incisors as well as in cheek teeth of adult horses (AG 2 and AG 3) was approximately 10-fold higher than that in foals (AG 1), with highest values in the subocclusal level in the peripheral zone 1.a (AG 2: incisors: m = 10.67/2000 µm^2^, cheek teeth: m = 27/2000 µm^2^).

Regarding the assessed horizontal subocclusal (*so*), central (*c*), and apical (*a*) levels, nerve fiber density showed a decreasing trend from subocclusal to apical in AG 1 and AG 2 in incisors as well as in cheek teeth (cheek teeth: AG 2: *so* vs. *a*: *p* = 0.03; see Appendix A). This decreasing trend was also obtained in cheek teeth of elderly horses (AG 3: within zone 3: *so* vs. *c*: *p* < 0.001; *so* vs. *a*: *p* < 0.001) but not in their incisors. In incisors of AG 3, the amount of nerve fibers remained almost constant regarding the different assessed horizontal levels (*so*, *c*, *a*). Regarding the defined zones, the highest density of nerve fibers was recorded in the peripheral zones 1.a, 1.b, and 1 in incisors as well as in cheek teeth, showing constantly lower values or a decreasing trend from zones 1 to 4, i.e., from peripheral to central (cheek teeth: AG 2: zone 1.a vs. 4: *p* = 0.03; see Appendix A). In general, the amount of nerve fibers per 2000 µm^2^ in cheek teeth was higher than that in incisors, with absolute highest values in zone 1a for AG 2 (m = 20.11/2000 µm^2^) and AG 3 (m = 19.38/2000 µm^2^).

### 3.3. Fibroblastic Cells

In incisors, the density of fibroblastic cell nuclei per 2000 µm^2^ remained almost constant from AG 1 to AG 3 (zone 1.a: AG 1: m = 28.5/2000 µm^2^; AG 2: m = 30.3/2000 µm^2^; AG 3: m = 26.22/2000 µm^2^; see Figure 6 and Figure 7), showing no statistical significances. In contrast to the finding in incisors, the density of fibroblastic cell nuclei in cheek teeth significantly increased from AG 1 to AG 3 (AG 1 vs. AG 3: *p* ≤ 0.05 (see Appendix A); zone 1.a: AG 1: m = 19.18/2000 µm^2^; AG 2: m = 24.67/2000 µm^2^; AG 3: m = 73.89/2000 µm^2^). In incisors of all age groups (AG 1–AG 3) as well as in cheek teeth of foals and adult horses (AG 1, AG 2), a significant decrease in the number of fibroblastic cell nuclei per 2000 µm^2^ was obtained from subocclusal (*so*) to central (*c*) and to apical (*a*) (incisors: AG 2: *so* vs. *c*: *p* = 0.028; *so* vs. *a*: *p* = 0.008; cheek teeth: AG 1: *so* vs. *a*: *p* = 0.014; see Appendix A). In contrast, cheek teeth of elderly horses (AG 3) showed an increasing trend from *so* to *a*, with low values at the central level and a high standard deviation.

Generally, the differences regarding the horizontal levels *so*, *c*, and *a* in incisors and cheek teeth were best visible in the peripheral zones 1.a, 1.b, and 1. Within the central zones 2, 3, and 4, the values remained almost constant throughout the age groups in incisors as well as in cheek teeth. Regarding the defined zones, the highest density of fibroblastic cell nuclei was recorded in the peripheral zones 1.a, 1.b, and 1 in incisors as well as in cheek teeth (except for AG 3), showing a significant and step-like decrease from zones 1 to 4, i.e., from peripheral in central direction (incisors: AG 2: zones 1.a and 1.b vs. zones 2, 3 and 4: *p* ≤ 0.01; zone 1 vs. zones 2, 3 and 4: *p* ≤ 0.05; cheek teeth: AG 2: zone 1.a vs. zone 1.b: *p* = 0.026; zone 1.a vs. zones 2, 3 and 4: *p* = 0.002; see Appendix A).

## 4. Discussion

The results of the present study suggest equine-specific histological characteristics and stratigraphic arrangements of relevant components such as blood vessels, nerve fibers, and cells of the dental pulp. Thus, the results provide novel data to enhance our understanding of the equine capacity for the prolonged production of large amounts of secondary dentine. Such knowledge might serve as a basis for further investigations to establish endodontic therapies adapted to the equine dentition.

### 4.1. Blood Vessels

Regarding the different age groups of the examined horses, the density of blood vessels showed an increasing trend with age in the equine dental pulp. This finding might be the effect of the permanent accumulation of newly formed secondary dentin at the inner walls of the pulp cavity, leading to a decrease in the size of the pulp cavity (Figure 8) [27]. However, it cannot be excluded by our data that the total number of blood vessels decreased while the cross-sectional area decreased in such an amount which caused a relative increase in blood vascular density. In contrast, a steady decrease in blood vessel number and diameter occurs with age in human brachydont teeth [28]. This phenomenon is explained by arteriosclerotic changes in the human dental pulp, resulting in a decreasing blood supply and a lower ability of defense mechanisms, such as tertiary dentin production, especially in the subodontoblastic layer [29]. The significantly higher vascularity of human deciduous teeth compared to permanent teeth is explained by the higher demand for the production of dental hard substances [30]. Remarkably, teeth of aged horses possess a similar or even higher vascularity than deciduous and/or younger teeth, which indicates a high prolonged productivity.

Our findings concerning the density of small blood vessels with a diameter < 10 µm in incisors along with cheek teeth show the highest density in the peripheral zones 1.a, 1.b, and 1, next to the odontoblastic layer. Blood vessels with a diameter up to 50 µm or even larger were not obtained in these peripheral zones but in the more central parts of the dental pulp. These results are not surprising as it is well known that for brachydont teeth of humans and dogs, large (>50 µm) and middle-sized blood vessels (up to 50 µm) gain access through the apical foramina, run in the occlusal direction in the central part of the dental pulp, and then branch out in the peripheral directions [31,32]. Similar to brachydont teeth, the accumulation of small blood vessels with a diameter up to 10 µm in the subodontoblastic region of the equine dental pulp proposes the presence of a nutritive subodontoblastic capillary network [33,34,35]. Furthermore, in equine pulp horns of permanent teeth (AG 2 and AG 3), some blood vessels of the subodontoblastic capillary network branch into the odontoblastic layer and were seen embedded between the odontoblasts, but they did not reach the predentin layer. This result is almost in line with reports for brachydont teeth, where capillaries also pass through the odontoblastic layer [22]. However, in brachydont teeth, these capillaries are reported to form a supportive network adjacent to the predentin [21], which has not been obtained in the horse. Due to the presence of a high density of capillaries in the subodontoblastic area, the dental pulp is likely able to quickly react on any inflammation with a higher blood flow, having a better capacity for tertiary dentine production [36,37].

### 4.2. Nerve Fibers

In deciduous incisors and cheek teeth of foals (AG 1), the lowest density of nerve fibers was recorded, and a more or less uniform distribution within zones 1.a to 4 was detected. The data obtained do not confirm the existence of a plexus-like arrangement of nerve fibers in primary teeth as there is a trend but no clear accumulation in the peripheral zones within the subodontoblastic area. In contrast to the findings in AG 1, in permanent incisors and cheek teeth of horses in AG 2 and AG 3, an accumulation of nerve fiber density in zone 1 was observed. Furthermore, numerous branching of nerve fibers in the peripheral zones 1.a, 1.b, and 1 was recorded, especially within the subocclusal level. This result suggests the development of a subodontoblastic nerve-fiber plexus (referred to as “Rashkow-plexus”) [19] in the equine dental pulp but not a constant existence in all ages of the equine tooth. This finding reflects the anatomical condition within brachydont teeth of humans and dogs as the Rashkow-plexus is not established in primary teeth until root formation is complete [38,39]. In human teeth, fibers from this subodontoblastic nerve plexus branch out into the dentinal tubules for about 50 µm, following the processes of the odontoblasts [40,41]. In equine teeth, this morphological characteristic is not established, presumably because dentin is exposed at the occlusal surface caused by physiological dental wear. 

Regarding the investigated horizontal levels, a decrease in nerve fiber density was obtained in apical direction, except for cheek teeth of AG 3. A similar nerve fiber distribution has been reported in brachydont teeth, where the plexus-like arrangement is supplied by larger nerve bundles and trunks, running parallel and without branching in apical direction [42,43]. The constant decrease in nerve fiber density from subocclusal to apical in AG 2 was not obtained in AG 3. In incisors and cheek teeth of older horses (AG 3), a rather irregular distribution of nerve fibers was obtained. For brachydont teeth, a continuous decrease in nerve fiber density has been reported [44,45]. In contrast, the equine dental pulp features a more or less constant density of pulpal nerve fibers. This might be explained by the compensating production and deposition of secondary dentin on the inner walls of the pulp cavity, which leads to a reduction in the size of the pulp cavity and to a shrinkage of the pulpal tissue [46].

### 4.3. Fibroblastic Cell Nuclei

In equine incisors of all age groups, the density of fibroblastic cell nuclei remains almost constant, whereas equine cheek teeth show a significant increase with age. In contrast, in brachydont teeth, cell density decreased significantly from the age of 20–40 years to 60–80 years by as much as nearly 60%, resulting in a reduced capacity of pulpal repair [47,48,49,50]. This age-related increase in cell density in the pulp of equine cheek teeth presumably reflects a high proliferative activity of the pulpal cells. However, it might be, at least partly, explained by the continuous shrinkage of the pulp tissue, caused by the lifelong deposition of secondary dentine at the inner walls of the pulp cavity [29,51]. 

Equine incisors as well as cheek teeth showed the highest density of fibroblastic cell nuclei in the peripheral zones 1.a, 1.b, and 1, similar to the distribution pattern of small blood vessels (< 10 µm) and nerve fibers. However, a cell-free zone was not observed in equine teeth, as defined for brachydont teeth [11,23]. Such a cell-free zone is frequently described for brachydont teeth, lying immediately beneath the odontoblastic layer [12] and having a width of about 40 µm [10]. However, the existence of a cell-free zone in brachydont teeth is still under debate [12,14,15,16]. Some authors suggest that a clear stratigraphic zonation is correlated with a decreased capacity for dentin production and, therefore, a common finding in the brachydont crown pulp but absent from the root pulp [8,20]. This hypothesis is supported by our findings as a zonation, in terms of cell-free and cell-rich zones, was not obtained in the highly productive equine dental pulp.

## 5. Conclusions

The equine dental pulp features a stratigraphic morphology which differs from the pulp of brachydont teeth. The equine specific pulpal morphology reflects the particular requirement for a prolonged high productivity of secondary dentine. Therefore, we suppose to define a subodontoblastic supportive zone, showing a high cellularity accompanied by high densities of blood vessels and nerve fibers. Such a supportive zone is present even in old horses and suggests that the dental pulp of equine teeth remains in a juvenile and active status, showing a high capacity of dentine production. In the future, this remarkable potential might become utilized specifically for endodontic and tooth-preserving therapies in equine dental medicine, which can—at least partly—replace dental extractions.

## 6. Limitations

As the horses were euthanized for other reasons than dental diseases, a systematic selection of specimens with favored age, sex, and breed was not possible. Furthermore, the number of horses and teeth had to be kept low to allow a broad methodical spectrum and investigations of several parts of the tooth. The irregular shape as well as the varying size of pulpal histological sections did not allow for uniform, standardized, and shape-matching areas to be defined for morphometric measurements. This might have caused biased results. Furthermore, it must be considered that angular deviations in the histological cross-sections produced sections of blood vessels displaying incorrectly enlarged diameters. The authors are aware that no statement can be made about the cellular activity as this was a post mortem cross-sectional study.

## Figures and Tables

**Figure 1 vetsci-09-00602-f001:**
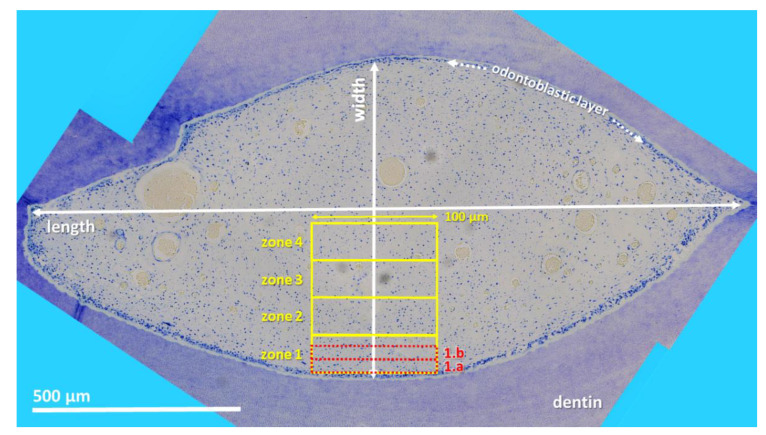
Cheek tooth (108) from a 21-year-old horse, toluidine blue-stained horizontal histological section (central level). Using LAS X, XY Live Image Builder, 80–100 single high-resolution images were recorded. For histomorphological purposes zones were defined. Zones 1–4 (yellow rectangle): constant length (100 µm) and recalculated width. Zones 1.a and 1.b (red rectangle): constant length (100 µm) and width (20 µm).

**Figure 2 vetsci-09-00602-f002:**
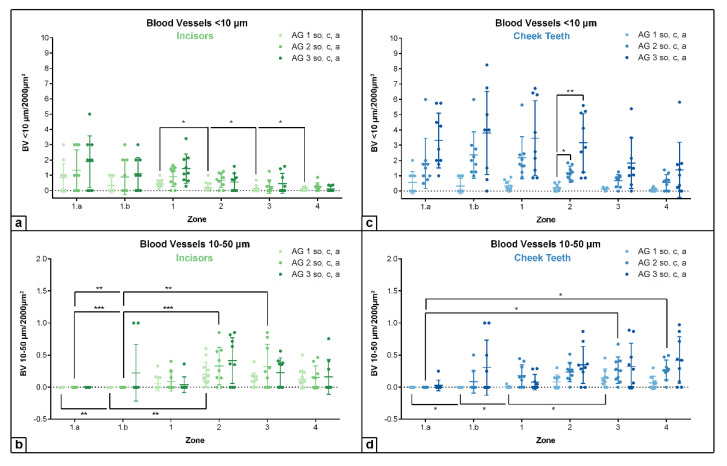
Dot-plot diagram showing the density of blood vessels (number per 2000 µm^2^) with different diameters (<10 µm, 10–50 µm) of different age groups (AG 1 to 3) in different zones (zone 1.a to 4). Mean was presented as a central horizontal bar and the standard deviation as vertical whiskers. *p*-values: *p* ≤ 0.05 (*), *p* ≤ 0.01 (**), and *p* ≤ 0.001 (***). (**a**) Dot-plot diagram showing the density of blood vessels < 10 µm of incisors (green colors). (**b**) Dot-plot diagram showing the density of blood vessels 10–50 µm of incisors (green colors). (**c**) Dot-plot diagram showing the density of blood vessels < 10 µm of cheek teeth (blue colors). (**d**) Dot-plot diagram showing the density of blood vessels 10–50 µm of cheek teeth (blue colors).

**Figure 3 vetsci-09-00602-f003:**
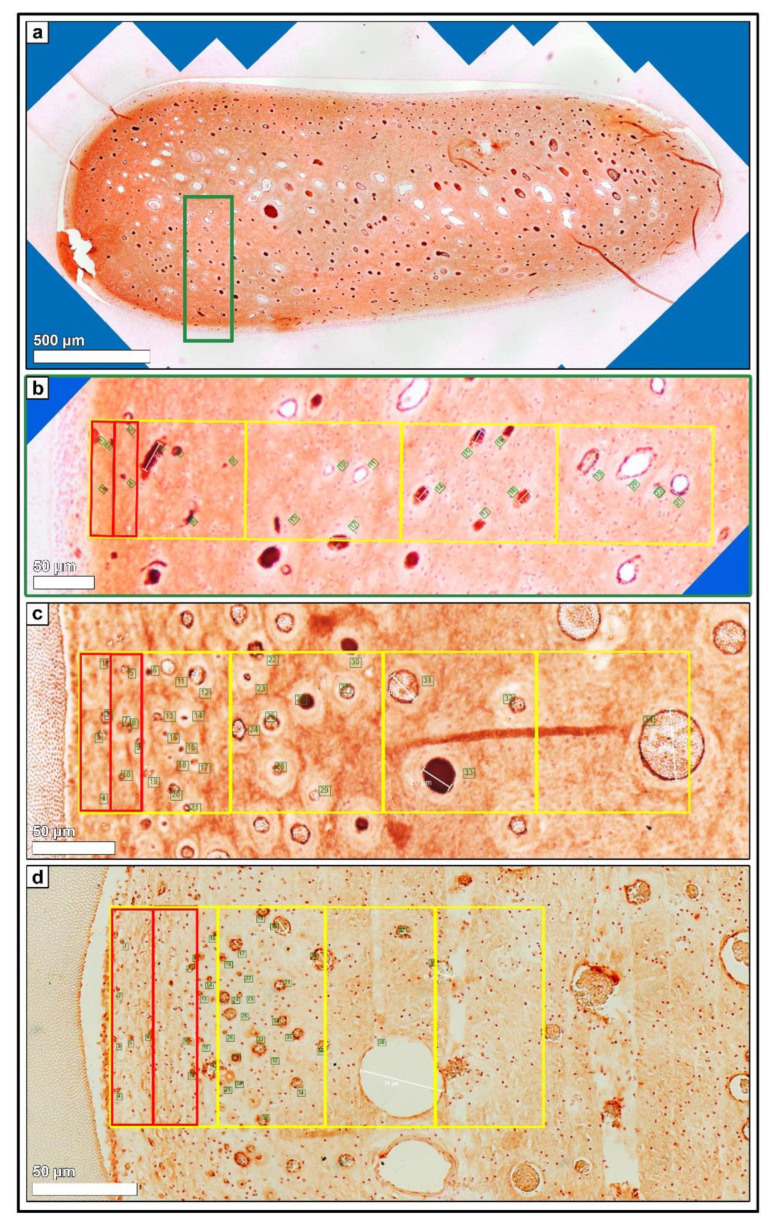
(**a**) Horizontal histological section (*so*) of a deciduous incisor (801) of a foal (210 days-old), immunohistochemical stain (blood endothelial cells, vWF VIII). 80–100 single high-resolution images were recorded using LAS X, XY Live Image Builder. (**b**) Green rectangle, detailed high-resolution image (vWF VIII). Zones 1.a and 1.b (red rectangle) and zones 1 to 4 (yellow rectangle) were indicated. Green numbers mark the counted blood vessels. (**c**) Detailed high-resolution image (vWF VIII) of a cheek tooth (208) of an adult horse (14 years-old). Zones 1.a and 1.b (red rectangle) and zones 1 to 4 (yellow rectangle) were indicated. Green numbers mark the counted blood vessels. (**d**) Detailed high-resolution image (vWF VIII) of a cheek tooth (108) of an adult horse (21 years-old). Zones 1.a and 1.b (red rectangle) and zones 1 to 4 (yellow rectangle) were indicated. Green numbers mark the counted blood vessels.

**Figure 4 vetsci-09-00602-f004:**
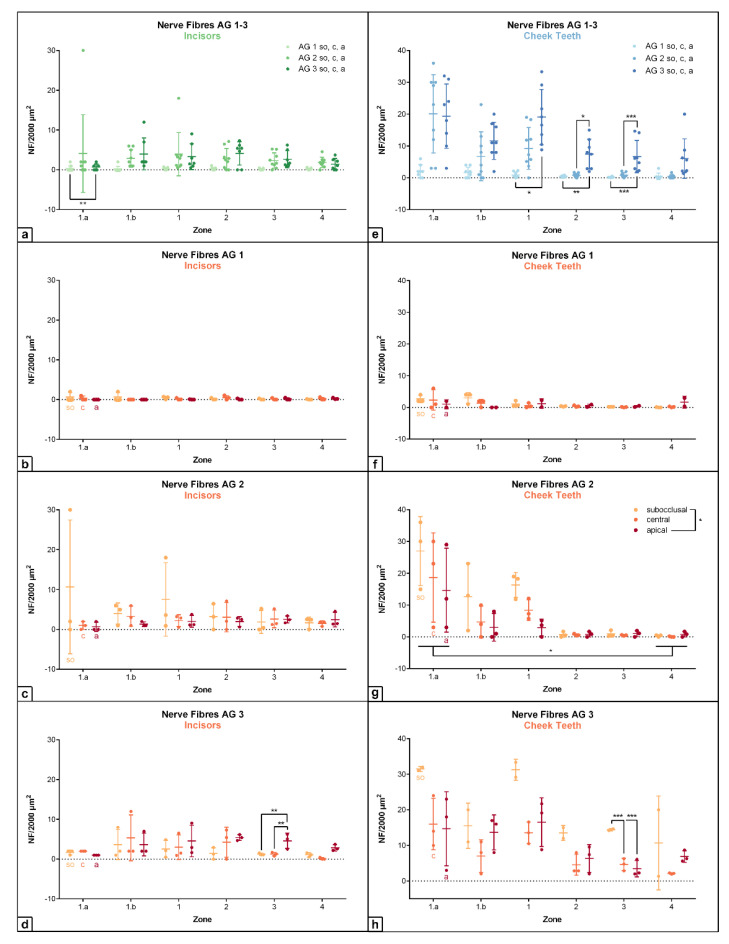
Dot-plot diagram showing the density of nerve fibers (number per 2000µm^2^) of different age groups (AG 1 to 3) within different zones (zone 1.a to 4). The horizontal levels subocclusal (*so*), central (*c*) and apical (*a*) are shown within the different age groups. Mean was presented as a central horizontal bar and the standard deviation as vertical whiskers. *p*-values: *p* ≤ 0.05 (*), *p* ≤ 0.01 (**), and *p* ≤ 0.001 (***). (**a**) Dot-plot diagram showing the density of nerve fibers in incisors of AG 1, AG 2 and AG 3 (green colors). Horizontal levels (*so*, *c*, *a*) are combined within each age group. (**b**) Dot-plot diagram showing the density of nerve fibers in incisors of AG 1 (orange colors). (**c**) Dot-plot diagram showing the density of nerve fibers in incisors of AG 2 (orange colors). (**d**) Dot-plot diagram showing the density of nerve fibers in incisors of AG 3 (orange colors). (**e**) Dot-plot diagram showing the density of nerve fibers in cheek teeth of AG 1, AG 2, and AG 3 (blue colors). Horizontal levels (*so*, *c*, *a*) are combined within each age group. (**f**) Dot-plot diagram showing the density of nerve fibers in cheek teeth of AG 1 (orange colors). (**g**) Dot-plot diagram showing the density of nerve fibers in cheek teeth of AG 2 (orange colors). (**h**) Dot-plot diagram showing the density of nerve fibers in cheek teeth of AG 3 (orange colors).

**Figure 5 vetsci-09-00602-f005:**
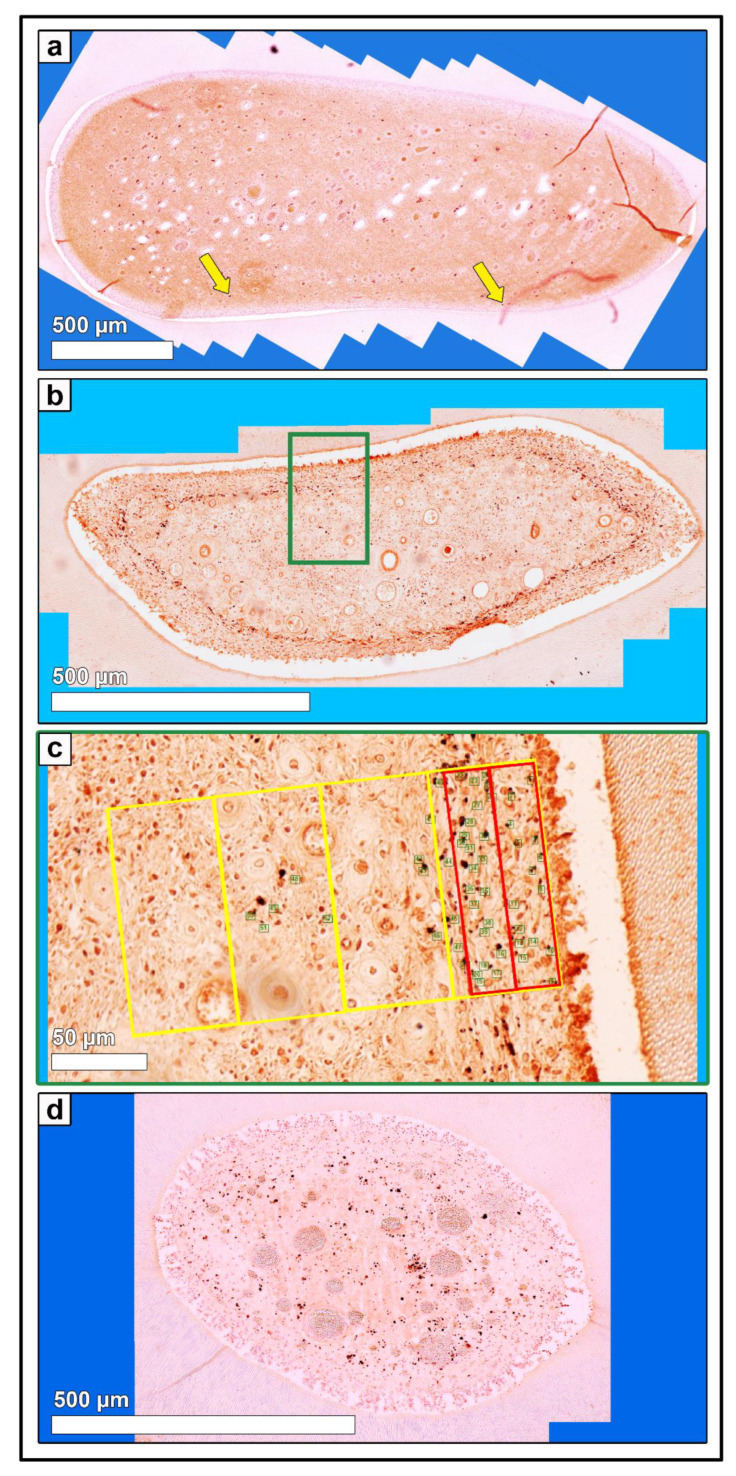
(**a**) Horizontal histological section (*so*) of a deciduous incisor (801) of a foal (210 days-old, AG 1), immunohistochemical stain (neurofilament protein, NF). 80–100 single high-resolution images were recorded using LAS X, XY Live Image Builder. Yellow arrows: indicate exemplarily isolated nerve fibers in the subodontoblastic area. (**b**) Horizontal histological section (*so*) of a permanent cheek tooth (208) of an adult horse (12 years-old, AG 2), immunohistochemical stain (NF). In the peripheral area of the dental pulp a subodontoblastic nerve fiber plexus is visible (accumulation of dark brown stainings). (**c**) Green rectangle, detailed high-resolution image (NF). Zones 1.a and 1.b (red rectangle) and zones 1 to 4 (yellow rectangle) were indicated. Green numbers mark the counted nerve fibers. (**d**) Horizontal histological section (*a*) of an incisor (301) of an older horse (19 years-old, AG 3), immunohistochemical stain (NF). In the peripheral area of the dental pulp a subodontoblastic nerve fiber plexus is visible (accumulation of dark brown stainings).

**Figure 6 vetsci-09-00602-f006:**
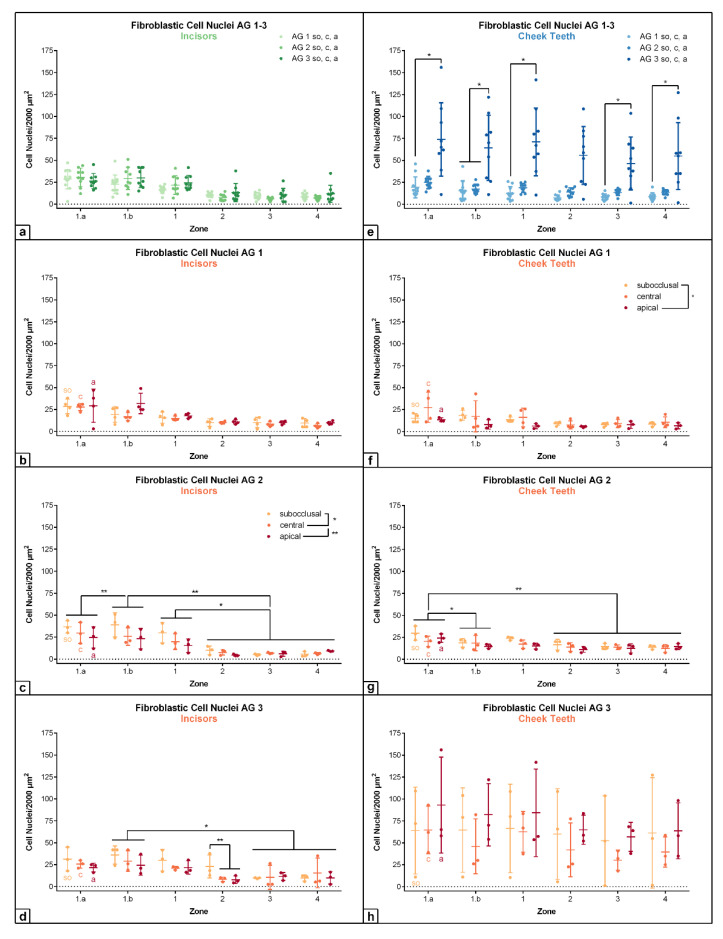
Dot-plot diagram showing the density of fibroblastic cell nuclei (number per 2000 µm^2^) of different age groups (AG 1 to 3) within different zones (zone 1.a to 4). The horizontal levels subocclusal (*so*), central (*c*) and apical (*a*) are shown within the different age groups. Mean was presented as a central horizontal bar and the standard deviation as vertical whiskers. *p*-values: *p* ≤ 0.05 (*) and *p* ≤ 0.01 (**). (**a**) Dot-plot diagram showing the density of fibroblastic cell nuclei in incisors of AG 1, AG 2 and AG 3 (green colors). Horizontal levels (*so*, *c*, *a*) are combined within each age group. (**b**) Dot-plot diagram showing the density of fibroblastic cell nuclei in incisors of AG 1 (orange colors). (**c**) Dot-plot diagram showing the density of fibroblastic cell nuclei in incisors of AG 2 (orange colors). (**d**) Dot-plot diagram showing the density of fibroblastic cell nuclei in incisors of AG 3 (orange colors). (**e**) Dot-plot diagram showing the density of fibroblastic cell nuclei in cheek teeth of AG 1, AG 2 and AG 3 (blue colors). Horizontal levels (*so*, *c*, *a*) are combined within each age group. (**f**) Dot-plot diagram showing the density of fibroblastic cell nuclei in cheek teeth of AG 1 (orange colors). (**g**) Dot-plot diagram showing the density of fibroblastic cell nuclei in cheek teeth of AG 2 (orange colors). (**h**) Dot-plot diagram showing the density of fibroblastic cell nuclei in cheek teeth of AG 3 (orange colors).

**Figure 7 vetsci-09-00602-f007:**
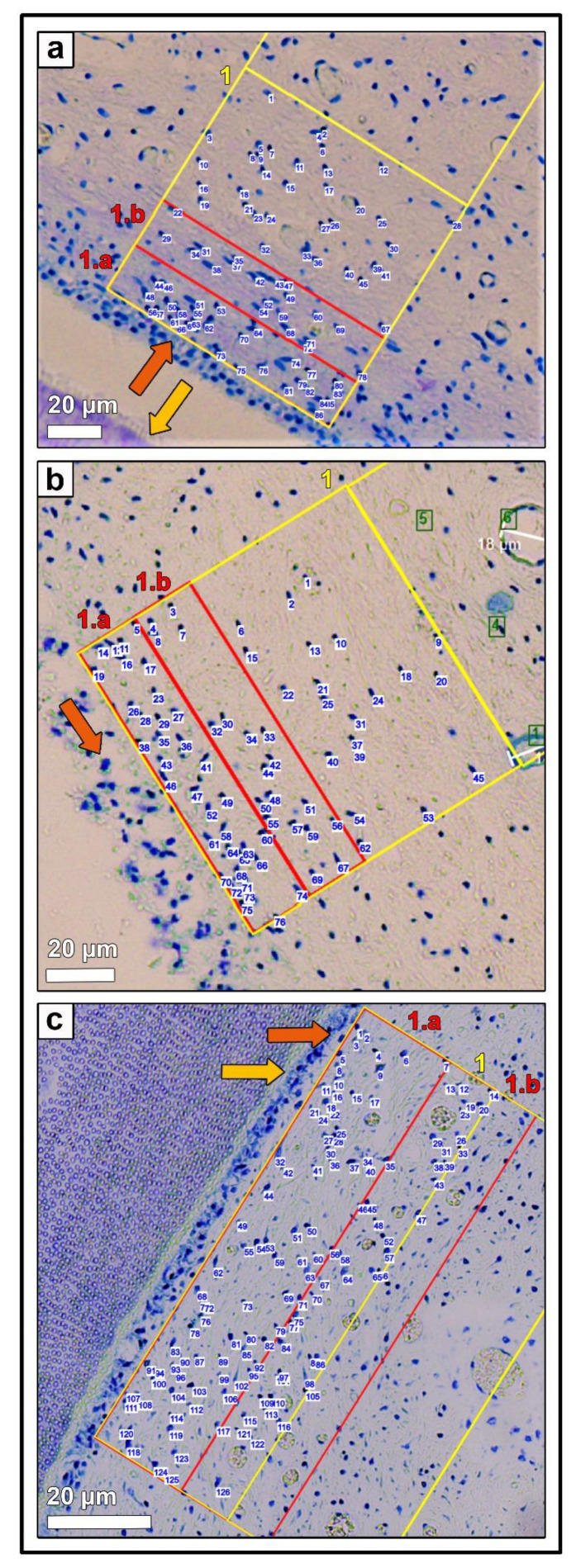
(**a**) Horizontal histological section (c) of a deciduous incisor (801) of a foal (210 days-old, AG 1), toluidine blue stain. Zones 1.a and 1.b (red rectangle) and zone 1 (yellow rectangle) are indicated, orange arrow: odontoblastic layer, yellow arrow: predentin layer. Blue numbers mark the counted fibroblastic cell nuclei. (**b**) Horizontal histological section (c) of a permanent incisor (401) of an adult horse (12 years-old, AG 2), toluidine blue stain. Zones 1.a and 1.b (red rectangle) and zone 1 (yellow rectangle) are indicated, orange arrow: odontoblastic layer. Blue numbers mark the counted fibroblastic cell nuclei. (**c**) Horizontal histological section (c) of a permanent cheek tooth (108) of an adult horse (21 years-old, AG 3), toluidine blue stain. Zones 1.a and 1.b (red rectangle) and zones 1 (yellow rectangle) were indicated, orange arrow: odontoblastic layer, yellow arrow: predentin layer. Blue numbers mark the counted fibroblastic cell nuclei.

**Figure 8 vetsci-09-00602-f008:**
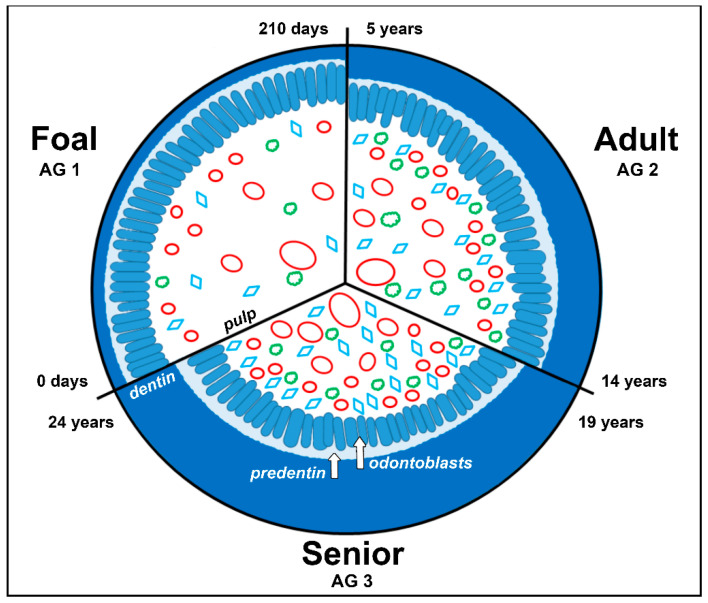
Schematic illustration of a decreasing pulp cavity of an equine tooth due to continued deposition of secondary dentin throughout age (0 to 24 years). Red ovals in different sizes: blood vessels, green clouds: nerve fibers, blue rhombus: fibroblastic cells.

**Table 1 vetsci-09-00602-t001:** Data of sampled horses.

No.	Sex	Breed	Age (d/y)	Age Group	Tooth	Reason for Euthanasia
1	mare	warmblood	2 d pre-part.	1	801	abortion
2	mare	blackforest draft horse	2 d	1	608, 701	colic
3	stallion	warmblood	5 d	1	608	septicemia
4	mare	warmblood	40 d	1	608, 801	colic
5	mare	shetland pony	210 d	1	608, 801	colic
6	mare	pony	5 y	2	208, 301	dystocia
7	gelding	warmblood	12 y	2	208, 401	colic
8	mare	warmblood	14 y	2	208, 401	atypical myopathy
9	mare	Icelandic horse	19 y	3	208, 301	colic, septicemia
10	mare	warmblood	19 y	3	401	colic
11	mare	warmblood	21 y	3	108, 401	colic
12	gelding	pony	24 y	3	108	laminitis

## Data Availability

The data presented in this study are available from the corresponding author upon request.

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
