# Peer review of "The Equine Dental Pulp: Analysis of the Stratigraphic Arrangement of the Equine Dental Pulp in Incisors and Cheek Teeth"

_vetsci, 2022, doi:10.3390/vetsci9110602_

Round 1

Reviewer 1 Report

The question that the authors address in this research paper is new, and is clearly specified in the introduction: “Can a stratigraphic organization of dental pulpa be observed in the equine hypsodont teeth as it is the case in the mature but not immature brachydont teeth as in humans?”. The experimental design in which different teeth of horses of various ages are sampled at different levels, with subdividing the pulpa in different zones along the central to peripheral axis and determining the density of structures such as blood vessels, nerve fibers and fibroblast is a sound and legit approach to this question.

Nevertheless, at several points, I had the impression that the authors have worked or reported to hastily and cut corners, necessitating either clarification or revision of the presented content. For the sake of ease, I will exemplify this impression by going chronologically through the text. The order below is therefore not a ranking of severity of my comments.

L47: as in previous lines the genuine subodontoblastic cell (free & rich) layers have been described, and also this paragraph and sentence starts with “subodontoblastic”, for the sake of clarity, it might be better to mention that Raschkow’s plexus is located even more centrally to the cell rich layer (so only per definition subodontoblastic, but not directly beneath the odontoblasts).

L57: It might be correct that little to no investigation has been done into the stratigraphy of the equine hypsodont dentition, however, I’m pretty sure that such research has already been performed in other hypsodont teeth, e.g. especially in the enigmatic narwhal tooth, and the nerve fibers it contains. It might be interesting, not compulsory, to refer to appropriate work in that field.

L64: sampling was only performed in the first upper incisor (I1 or Id1) and fourth lower premolar (P4 or Pd4). Although there is reason to believe that results taken from 1 incisor is applicable to all incisors, and idem for 1 premolar to all (maybe but P1) premolars and molars, it is nowhere clearly stated or proven that way. Nevertheless, in all subsequent paragraphs, the authors discuss the data as obtained from incisors and cheek teeth (in general). A little bit more caution is mandatory.

Table 1: Even if the animals have died of reasons unrelated to the study, the reason why they died still may have influenced the results. Both colics and septicemia have effects on general circulation and therefore can have had impact on the diameter of the blood vessels.

L78: The three defined horizontal levels are absolutely not well-defined. Just calling them subocclusal, central and apical isn’t sufficient, especially when discriminating between the two latter in aging horses. Moreover, the term central is a little ill-chosen (central in facts means also deep within (in contrast to peripheral), not just in the middle), and it is clear that the authors in their preparation of this work have struggled with the same problem, as sometimes this central part in the remainder of the text isn’t abbreviated with the letter c, but with the letter m (from middle?), as in line 222 “so vs m” and line 297 “histological section (m)”, in the supplemental data,… . Apart from all that, and for clear comparison & reference, the term central must be clearly defined. In my mind, a “central” part of a tooth is the vast, robust part, still a certain distance away from the division into different roots (in the premolar). To my surprise, the figure 01, displaying a horizontal cross section at the central level of the premolar did not meet my expectation: it is simple in shape, clearly one of the roots (pulp horns) that has been cross-sectioned,…until I saw the age group: old age, and therefore indeed worn teeth… Due to wear, the geometric middle of the remaining tooth gradually shifts towards the root. Question is, for this research, and again in order not to compare apples with oranges, whether it is desirable to use the geometric middle of the remaining tooth as “central” part (consequentially, in an old horse, you sample more apically than in a young horse), or to keep the original length of the unworn tooth as reference to determine the central part. It is obvious that for the subocclusal region, one has to take in account the tooth wear, as one wants to see what happens almost directly below the surface in the structures that risk to get exposed by wear. But for the central part: is it important to either (1) sample at a certain fixed distance from the occlusal surface, (2) at a relative distance between the occlusal surface and tooth apex, or (3) at a fixed spot in the tooth that gradually comes closer to the occlusal surface, but where sampling, regardless of age and wear, is done from the same tissue (at the same original level). For this research, I favor option 3, it’s up to the authors to convince me from the opposite.

L105: I see that no counterstaining has been perform for the immunohistochemical slides (e.g. with hematoxylin to visualize the cell nuclei). This means that all contrast on the slides is a result of the immunohistochemical reaction, either specific or not. Tissue components are remarkable well discernable on the provided slide pictures, not only the ones that are expected to contain the vWF…, which is suggestive for a lot of background staining. Alas, the negative control to truly mark that the stronger contrast as seen in presumptive blood vessel walls is a specific stain is missing. The authors did omit the primary antibody as negative control, but failed to include negative controls omitting both the primary and the secondary (peroxidase-linked) antibody, to genuinely prove that the quenching they performed was successful and no intrinsic peroxidase reaction is to blame for higher contrast/staining observed around blood vessels when only the AEC is applied.

L111:”radial symmetry”: a complex cross-section of a tooth cannot be radial symmetric. Reformulate (“radial relative homogeneous distribution of tissue component & cell types”?). Apart from that, the assessment of this homogeneous distribution relative to a percentage of the central to peripheral axis is an important item to justify the selection of only one, single-sided set of 4 zones as reference for the entire cross section. The authors should elaborate more on this assessment: how did you verify homogeneity at all levels?

L118: only zones 1.a and 1.b are exactly 2000 square µm. The sizes of all 4 zones are relative. How did you assess density (authors mention counting numbers, but results are densities /2000 µm²): counting every vessel/nerve fiber in a zone, calculate the entire area and recalculate as density per 2000 µm², of choose a random area or multiple areas within the zone that add up to 2000µm² and count numbers in these areas, or count in areas of less than 2000 µm² and multiply….?? Furthermore, in density assessments, certain rules apply in order to obtain unbiased results. Sectioning, placing of the zones on top of the section,… must have been performed in a standard randomized way. What rules were applied if vessel signatures crossed border lines of the zones or areas to count (when were they counted in, when not?): all of this must be thoroughly described and of course executed in a correct way.
In standard random sampling, the set of 4 adjacent zones should normally have been randomly plotted on the cross section (along any of the radials). This means that a randomly dropped counting area can fall on more acutely or irregularly shaped borders of the tooth, so also e.g. to the left or the right of figure 1…. As the authors are interested in the question of stratigraphy, the borders of the zones should then equidistantly follow the line of odontoblasts, and display a more complex geometry, rather than consist of straight lines (also to adhere to the observed “radial” symmetry). I assume that the authors however did not perform random sampling, and looked for an area with the most linear / straight arrangement of odontoblast to construct their set of  zones, and therefore, sampling was biased.

L123: regarding the blood vessel diameter measurements. Little is said about the applied measuring technique, but some figures, e.g. figure 3c, contain some measurements (not mentioned in the caption) that let me fear the worst regarding a correct and standardized assessment. For instance, the larger, darkest stained vessel in fig 3c: the measurement displayed here is not a diameter, rather a chord of the circle the vessel describes… The vessel that is crossing the right border of zone 4 is measured along its widest diameter, whilst it is clear that this vessel doesn’t describe a perfect circle, indicative for the fact that the vessel has been cut in an oblique way. In obliquely cut vessels, only the NARROWER diameter of the oval is representative for the true diameter, certainly not the widest diameter spotted, as this would overestimate the true diameter of the vessel. Circular vessels should have been measures at least 3 times in different directions, and the results averaged. In cases such as this one, where measuring is only necessary for rough subdivision of vessels, the problem of crude measurement is not that severe, however for borderline cases (less or more than 10 µm), it can make a difference.

L127: figure 1: who are the arrows “length” and “width” included? Treacherous, since the “length” does not correspond with the length of the tooth (horizontal cross-section).

L187: figure 2 (and also figures 4 & 6): textwork & graphs are typically read from left to right and then from top to bottom. I certainly do understand and support the arrangement of the graphs (left: all info on incisors, right: all info on the premolars), but maybe to indicate that arrangement and labelling (a-b-c-d,..) is different than expected, left and right graphs might be separated a little more, e.g. by a narrow white band, rather than just the black centerline between them.

L187: I needed the supplemental files to exactly figure out what datasets are statistically significantly different from one another. The graphical representation of the latter is poor and subject to misunderstanding (comment also applicable to figures 4 & 6).

L188: continuity of text, lines 192-195 should be placed first.

L188-191: mentioning of graph colors under the tags a-d is irrelevant. All line colors in a single graph (e.g.: a) are shades of the same color (in this case green), and hold no information. Colors can, but should not necessarily be referred to in the general caption of the graph (as there is a difference between the top graphs and the remainder), before individually discussing the subsets a-d (remark also valid for figures 4 & 6).

L226-227: take more care in formulating items and their statistical relevance. It is correct that only a decreasing trend was seen regarding NF density from zone 1 over zones 2 and 3 to zone 4: no statistical differences between each adjacent zones. The only statistical difference was between zones 1 and 4, as mentioned between brackets. However, these brackets can be interpreted as belonging to the preceding statement in the same sentence (referring to the trend mentioned).

L231 Figure 4: legend to the different orange shades is only given in figure 4f: either include it everywhere, or outside the boxes, as a general legend. The same comment is valid for figure 6 (6c and 6f contain legends to the orange shades, the rest not).

L266 & many other places in the text: replace “obtained” by “observed” (perform a search – find & replace action on the entire text, including figure captions).

L296: caption starts oddly in the right bottom area of the page

L310: Don’t use the word “thus” in scientific conclusions. Moreover, the statement in that one sentence is not a direct consequence of the previous sentence, so even an alternative word of “thus” is irrelevant.

L316: the statement in this sentence (surprising increase in density of blood vessels) is in clear contradiction with the observations mentioned in L160 (no statistical difference (only a trend), except in one single and very specific case). It is therefore a too bold conclusion.

L316-317: the assumption of, due to age, getting less space for a similar amount of content could have easily been tested by the authors by not only calculating the densities in zones with a specific area, but also determining the total cross-sectional area of the entire pulpa, so that the fraction of both can give a rough estimate of total number of (cross-sectioned) blood vessels and hence support the idea that crowding of blood vessels can mainly be attributed to the lesser space they get. This however doesn’t exclude the possibility that the number of blood vessels is decreasing (as observed in the brachydont teeth as explained in the next sentences! Relative densities can still rise if the rate of loss of available space for blood vessels surpasses the rate of loss of number of blood vessels. In other words, both observations (increase of relative density (as putatively observed in this study, see remark above) versus decrease in total number of blood vessels, as observed in the brachydont teeth (L319) can co-exist in a single, aging tooth, the one does not exclude the other. Authors should therefore tone down their level of surprise (unless they calculate total numbers by also assessing total pulpa area and make a fraction with relative densities).

L340: embedded capillaries in the odontoblast layer (please use embedding or protruding, rather than “isolated between”), although clearly described, are such an extraordinary feature (i.e. vessels (seemingly) be present inside an epithelium), that inclusion of a picture of this observation is mandatory. Although not the scope of the paper, even better, a TEM image (clearly depicting the fate of the basement membranes) might further support this statement made here.

L355-357: none of these observations was statistically significant (except for a small difference between zone 1 and 4 in AG2 premolars), therefore, one cannot conclude that there is a higher density of nerve fibers in certain layers justifying the presence of a Rashkow plexus.

L397: figure 8: omit the arrows (red and yellow), and make both your odontoblast layer and predentin layer stand out in a specific color all around. Refer to these layers by color. Sidenote: especially true for previous figures working with shades of orange and green: these colors are not helpful to the color-blind. Maybe use more distinctive and inclusive colors, and in the present figure 8, also use striping / patterns.

L412: the list of limitations should be completed with the comments above that cannot be resolved by the authors.

Author Response

Dear Reviewer 1,

thank you very much for your answers.

Please find attached the answers to your comments.

Yours sincerely,

Jessica Roßgardt

Reviewer 2 Report

Introduction

The introduction provides a sensible justification for why work was undertaken. Literature review is appropriate. A histological study describing the cell types present is perfectly justified, but if there are practical implications for the results of the work it would be useful to include them in the introduction.

Materials and methods

The authors state that “Subsequently, all samples were treated according to Roßgardt et al. [24].” It would be helpful to explain what the nature of the handling was. It is not necessary to go into details or procedures but rather to explain the general processes and their purposes.

Table 1 uses a triadan scale to identify teeth. The scale should be referenced as readers may not be familiar with the concept that deciduous teeth are numbered 5-8.

The handling of samples appears to be appropriate.

How subjective was the analysis of the histological specimens? How many people analysed each specimen? If only a single observer was used, was there repeat analysis of the specimens to assess reproducibility of results?

The statistical analysis is appropriate.

Results

The results are presented appropriately.

Discussion

The discussion is appropriate. It answers the questions that were asked in the introduction and the conclusions that are drawn are justified from the data presented.

Once again, if the authors can suggest practical outcomes related to their findings it would be useful to include them. It is certainly not essential but would add to the value of the manuscript for readers who were not specifically focused on histology.

Author Response

Dear Reviewer 2,

thank you very much for your answers.

Please find attached the answers to your comments.

Yours sincerely,

Jessica Roßgardt
